# Health-Related Quality of Life in Patients with Chronic Myeloid Leukemia Treated with First- Versus Second-Generation Tyrosine Kinase Inhibitors

**DOI:** 10.3390/jcm9113417

**Published:** 2020-10-25

**Authors:** Adi Shacham Abulafia, Sivan Shemesh, Lena Rosenmann, Tamar Berger, Avi Leader, Giora Sharf, Pia Raanani, Uri Rozovski

**Affiliations:** 1Institute of Hematology, Davidoff Cancer Center, Beilinson Hospital, Rabin Medical Center, Petah-Tikva 49100, Israel; lenaro1@clalit.org.il (L.R.); tamaritayberger@gmail.com (T.B.); avileader@yahoo.com (A.L.); praanani@012.net.il (P.R.); uriro1@clalit.org.il (U.R.); 2Sackler School of Medicine, Tel Aviv University, Tel Aviv 6997801, Israel; shesivan1@gmail.com; 3Israeli CML Patients Organization, 5 Ehud Manor st., Netanya 4265952, Israel; giora1@inter.net.il

**Keywords:** quality of life, chronic myeloid leukemia, tyrosine kinase inhibitors, patient-reported outcome

## Abstract

The life expectancy of patients with chronic myeloid leukemia (CML) approaches that of the age-matched population and quality of life (QOL) issues are becoming increasingly important. To describe patients’ characteristics and assess QOL, we delivered a 30-item core questionnaire, a 24-item CML-specific questionnaire, both from the European Organization for Research and Treatment of Cancer (EORTC), and additional health-related items to 350 patients. Among 193 patients who completed the questionnaires, 139 received either imatinib (*n* = 70, 33%), dasatinib (*n* = 45, 23%) or nilotinib (*n* = 24, 12%). Patients’ median age was 58 (range: 23 to 89) years and 86 (63%) were males. Stratifying patients by treatment, we recognized two distinct populations. In comparison to patients on dasatinib and nilotinib, patients on imatinib were two decades older, had a longer duration of disease and current treatment, experienced fewer limitations on daily activities (*p* = 0.02), less fatigue (*p* = 0.001), lower degree of impaired body image (*p* = 0.022) and less painful episodes (*p* = 0.014). Similarly, they had better emotional functioning, were less worried, stressed, depressed or nervous (*p* = 0.01) and were more satisfied with their treatment (*p* = 0.018). Not only does age associate with current treatments, but it also predicts how patients perceive QOL. Young patients express impaired QOL compared with elderly patients.

## 1. Introduction

In 2001, imatinib was approved for the treatment of patients with chronic myeloid leukemia (CML) [1]. Since then, several generations of newer, more potent tyrosine kinase inhibitors (TKIs) entered the market. As of 2020, five drugs have been approved in the US and Europe for CML. Yet, the EUTOS registry data indicate that three drugs are most commonly used. Patients in chronic-phase CML typically receive lifelong treatment with either imatinib, dasatinib or nilotinib [2]. Treatment with any of these drugs is associated with high survival expectancy. Nonetheless, they come with a price of adverse effects, where some are common to all TKIs and some are drug-specific.

The choice of TKI is tailored based on drug availability, patients and disease characteristics and whether treatment discontinuation is considered a goal [3]. Since, nowadays, the life expectancy of patients with CML approaches that of the general age-matched population [4,5], quality of life (QOL) issues are becoming increasingly important. QOL studies typically focused on tolerability which is only one aspect of life quality [3,6,7] and only limited data are available on the QOL of patients that are treated with these drugs outside clinical trials.

In an era of patient-centered medicine, patients-oriented research that brings out patients’ perspectives provides an additional layer of valuable knowledge. Using patients reported outcomes (PROs) unravels patients’ preferences and may ultimately leads to improved patient–physician communication, improved adherence to treatment and improved overall QOL.

In this study, we use PRO measures (PROMs) to assess the QOL and symptom burden in 139 patients with CML, treated with one of the three most commonly used TKIs—imatinib, dasatinib and nilotinib—outside clinical trials.

## 2. Material and Methods

This nationwide study was a joint initiative of the Israeli CML patients’ organization and the Institute of Hematology at the Rabin Medical Center, Israel. The study was approved by our local institutional review board (0034-19-RMC). The study included any adult patient (≥18 years old) with a diagnosis of CML who received at least one TKI for at least 3 months and agreed to complete self-reported questionnaires. We approached patients either by e-mails that were sent to members of the CML patients’ organization or by their treating physician while waiting for their appointment, if treated in our hospital. Patients who agreed to participate completed a survey booklet or a computerized version of it. The booklet was written in Hebrew and included the following questionnaires: a 30-item core questionnaire (QLQ-C30), and a 24-item CML-specific (QLQ-CML24) questionnaire, both provided by the European Organization for Research and Treatment of Cancer (EORTC). Additional questions included demographics and clinical data, and items involving anxiety, sexual and work function, family planning and adherence to treatment. A complete version of the protocol was added to the Appendix A.

*The QLQ-30* consists of (i) 5 functioning scales: physical, role, emotional, cognitive and social, (ii) 3 symptoms’ scales: fatigue, nausea/vomiting and pain, (iii) 6 single-item scales: dyspnea, insomnia, appetite loss, constipation, diarrhea and financial impact and (iv) the global health status quality of life scale [8]. The Hebrew version of this questionnaire was approved by the EORTC, and was already used in previous studies (https://www.eortc.org/). The alpha Cronbach test for internal consistency of the QLQ-30 subscales ranged between 0.69 and 0.92.

*The QLQ-CML24* consists of impact on daily life, symptom burden, impact on worry/mood, body image problems, satisfaction with care and information and satisfaction with social life. The items were scaled and scored according to the EORTC recommendations [9]. To validate the Hebrew version, we used the backward–forward translation strategy as previously described [10,11]. Briefly, 2 independent translators fluent in English and Hebrew translated the English version and the Hebrew version was re-translated to English. This process continued until the final version was approved by the EORTC committee. The alpha Cronbach test for internal consistency of the *QLQ-CML24* subscales ranged between 0.75 and 0.91.

All scales and single-item measures were standardized and the score ranges from 0 to 100. A high score for functional and QOL items/scales represents better function and QOL. A high score in symptoms items represents worse symptomatology.

We used the Mann–Whitney test to compare medians and χ2 to compare categorical variables.

## 3. Results

### 3.1. Patients’ Characteristics

Overall, 193 patients completed the questionnaires, the median age was 58 years (range: 23 to 89) and of those, 102 patients (53%) were males. Time from diagnosis ranged between less than a year to 46 years (median: 7 years). In this report, we included 139 patients (72%) who received either imatinib (*n* = 70, 33%), dastainib (*n* = 45, 23%) or nilotinib (*n* = 24, 12%) and did not include the few patients who received bosutinib (*n* = 8, 4%) or ponatinib (*n* = 2, 1%), patients who discontinued treatment (*n* = 22, 11%) or patients with missing data regarding the TKI they received (*n* = 22, 11%).

Stratifying patients by treatment, we recognized two distinct populations with unique demographics and clinical characteristics: patients on imatinib and those on second-generation TKIs (dasatinib or nilotinib). Patients on imatinib were on average two decades older (Table 1), had longer duration of disease and longer exposure to current drug (Table 2) in comparison to patients on dasatinib and nilotinib.

### 3.2. Symptom Burden-EORTC (The QLQ-30 and the QLQ-CML24)

The EORTC questionnaires included 34 items related to symptoms that were experienced by patients within the last week. These items were grouped into 13 symptom scales. Overall, the symptomatic profile was more severe in patients who received second-generation TKIs. This trend was more pronounced in patients on nilotinib and was independently shown in both the cancer-specific EORTC QLQ-30 and in the CML-specific CML-24 (Figure 1). For example, in the EORTC QLQ-30 questionnaire, compared with patients on either dasatinib or nilotinib, patients on imatinib experienced lesser limitations on daily activities (38 and 47 vs. 25, respectively, *p* = 0.02), less fatigue (53 and 57 vs. 37, respectively, *p* = 0.001) and a lower degree of impaired body image (38 and 46 vs. 26, respectively, *p* = 0.022). Similarly, in the QLQ-CML24 questionnaire, patients on imatinib reported less painful episodes compared with patients on nilotinib (28 vs. 47. *p* = 0.014). Furthermore, the overall symptom burden was higher with nilotinib (41) compared with either dasatinib (27, *p* = 0.005) or imatinib (30, *p* = 0.018).

### 3.3. Functional Status—EORTC (The QLQ-30 and the QLQ-CML24)

Additionally, the EORTC questionnaires include 18 items that are grouped into eight scales. These scales are aimed to estimate patients’ functional status. Compared with nilotinib, patients on imatinib had better emotional functioning and were less worried, stressed, depressed or nervous. The mean score of global emotional functioning was 77 for imatinib (standard deviation (S.D.) 24) and only 61 (S.D. 31) for nilotinib (*p* = 0.009) (Figure 2). Likewise, patients on imatinib were also more satisfied from the knowledge and treatment they received from their caregivers; the mean score on this scale was 41 (S.D. 37) for imatinib, 24 (S.D. 23) for dasatinib and only 19 (S.D. 26) for nilotinib (*p* = 0.01).

### 3.4. Additional Health-Related (HR) Items

In addition to the EORTC questionnaires, several items were added by us and are analyzed herein (Table 3).

Consistent with the results from the EORTC questionnaires, patients on imatinib reported better HR QOL during the last three months (*p* = 0.04) and more patients on imatinib reported that their overall QOL was not affected, or only mildly affected by the disease or treatment they received (*p* = 0.02).

Overall, as many as 44% (*n* = 58) reported that their function at work was moderately or severely impaired and 35% (*n* = 38) reported that their function at home was impaired. Yet again, only 33% (*n* = 22) of patients on imatinib reported significant work dysfunction compared with 56% (*n* = 24) and 52% (*n* = 12) of patients on dasatinib or nilotinib, respectively (*p* = 0.04) and likewise, only 24% (*n* = 16) of patients on imatinib reported significant home dysfunction compared with 47% (*n* = 21) and 46% (*n* = 11) of patients on dasatinib or nilotinib, respectively (*p* = 0.023). On the other hand, while 40% of patients (N = 51) reported that their sexual function deteriorated during treatment, there were no differences across drugs.

In the questionnaire, we also included several items that might point to drug toxicity including breathing difficulties due to pleural effusion, hospitalization during the previous six months and referral to cardiovascular or pulmonary clinics. We did not find any difference across drugs in any of these questions, nor in the reported adherence rates.

## 4. Discussion

In this “real-life” study, we used self-reported questionnaires to assess how patients perceive their QOL with imatinib, dasatinib and nilotinib, the three most common TKIs that are currently in use. Overall, we found that patients on imatinib expressed lower levels of distress and higher levels of satisfaction from the treatment that they received. Despite this, our study does not indicate that imatinib is a “better drug” or at least provides superior QOL.

Previous studies showed that age impacts health-related quality of life (HR QOL), but not necessarily in the expected direction. For example, in a pooled analysis of more than 6000 patients with cancer, social role and emotional functioning were better with increasing age [12]. Likewise, among 1420 women with breast cancer, HR QOL was better in older patients [13]. In line with these large-scale studies, in our study, patients on imatinib, who experienced improved HR QOL, were two decades older.

With a median age of 67, most patients on imatinib are “baby boomers”, born after World War II between 1946 and 1964. Compared with the “x generation” born between 1965 and 1976, boomers have a higher prevalence of “excellent” self-reported health [14]. Hence, differences between boomers, mostly on imatinib, and younger patients, mostly on second-generation TKIs, might reflect a sociological trend rather than “drug quality”. Moreover, a recent study that also used the EORTC questionnaires found that dasatinib provided a superior HR QOL profile compared with imatinib, although the advantage of dasatinib was more pronounced in the younger age group, and negligible after the age of 60 years [15].

That imatinib is primarily reserved for elderly patients and second-generation TKIs are given to younger patients reflects our current practice. Dasatinib and nilotinib are more potent drugs. Patients on dasatinib or nilotinib achieve earlier responses and the rate of deeper molecular responses, that define candidates for treatment discontinuation, are higher [16,17,18,19,20]. Clinical trials that compared imatinib and second-generation TKIs in different settings reported that the safety profile of imatinib and second-generation TKIs are comparable. Yet, patients on dasatinib and nilotinib experience higher rates of serious adverse events that put elderly patients with age-related health problems at particular risk. For example, approximately 30% of patients on dasatinib will experience pulmonary edema [19,21]. Likewise, nilotinib carries a black box warning in the United States for possible heart complications and affects electrolytes and glucose balance [20,22,23]. Therefore, in otherwise healthy younger patients, giving second-generation TKIs for a more limited duration is the preferred choice when treatment discontinuation is the ultimate goal.

In conclusion, approximately 15% of Israeli patients with CML participated in this “real-life” study. At the time of survey, we found mostly elderly patients on imatinib and younger patients on either dasatinib or nilotinib. Not only is age associated with current TKI treatment, but it also predicts how patients perceive their QOL. Young patients that face serious illness, typically for the first time, express higher levels of distress and impaired QOL when compared to elderly patients.

## Figures and Tables

**Figure 1 jcm-09-03417-f001:**
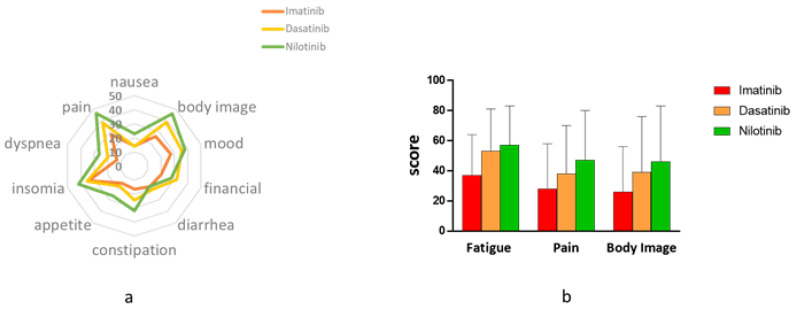
Symptom Burden on imatinib vs. dasatinib and nilotinib. Symptom burden according to EORTC QLQ-C30 and EORTC QLQ-CML24 in patients on imatinib, dasatinib or nilotinib. (**a**) A polar plot summarizes scores of 10 symptom-related items/scales. Higher scores represent worse symptomatology. As shown, patients on imatinib are represented in the inner circle, indicating less symptoms. (**b**) A bar graph showing that patients on imatinib report less fatigue and pain and a better body image. European Organization for Research and Treatment of Cancer Quality of Life Core Questioner 30 (EORTC QLQ-C30); European Organization for Research and Treatment of Cancer Quality of Life Questioner of Chronic myeloid Leukemia 24 (EORTC QLQ-CML24).

**Figure 2 jcm-09-03417-f002:**
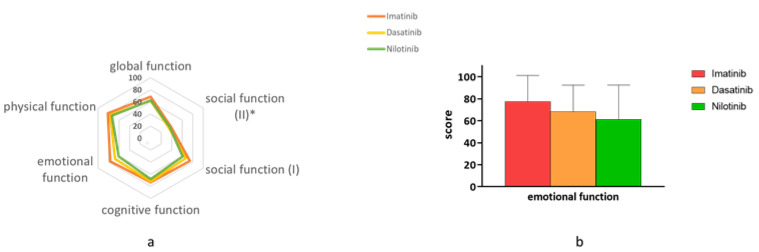
Functional Status on imatinib vs. dasatinib and nilotinib. Functional status according to QLQ30 and CML-24 in patients on imatinib, dasatinib or nilotinib. (**a**) A polar plot summarizes scores of 5 different aspects of function according to patient reports. Social function is represented in 2 scales (QLQ30 and CML-24). As shown, patients on imatinib are represented in the outer circle, indicating better function. (**b**) A bar graph showing that patients on imatinib report higher levels of emotional functioning. European Organization for Research and Treatment of Cancer Quality of Life Core Questioner 30 (EORTC QLQ-C30); European Organization for Research and Treatment of Cancer Quality of Life Questioner of Chronic myeloid Leukemia 24 (EORTC QLQ-CML24).

**Table 1 jcm-09-03417-t001:** Demographic characteristics of study population.

Characteristics	Imatinib *n* = 70	Dasatinib *n* = 45	Nilotinib *n* = 24	Total *n* = 139	*p* Value
Age, years, median (range)	67 (32 to 89)	47 (23 to 78)	50 (26 to 85)	58 (23 to 89)	<0.0001
Gender *n* (%) *					
male	43 (62)	28 (64)	15 (63)	86 (63)	0.99
female	26 (38)	16 (36)	9 (37)	51 (37)	
Family status *					
singles, *n* (%)	3 (5)	13 (30)	4 (17)	20 (15)	0.001
married, *n* (%)	62 (95)	30 (70)	20 (83)	112 (85)	
number of children, median (range)	3 (0 to 15)	2 (0 to 10)	2 (0 to 6)	3 (0 to 15)	0.06
Education level, *n* (%) *					
elementary	4 (6)	3 (7)	3 (12)	10 (7)	0.6
high school	24 (36)	19 (42)	6 (25)	49 (36)	
high education	39 (58)	23 (51)	15 (63)	77 (57)	
Level of religiosity, *n* (%) *					
secular	40 (62)	30 (68)	13 (57)	83 (63)	0.93
traditional	14 (22)	7 (16)	16 (26)	27 (21)	
religious	10 (15)	6 (14)	4 (17)	20 (15)	
ultra-orthodox	1 (1)	1 (2)	0	2 (1)	

* Data were not available for all patients.

**Table 2 jcm-09-03417-t002:** Disease characteristics of study population.

Characteristics	Imatinib *n* = 70	Dasatinib *n* = 45	Nilotinib *n* = 24	Total *n* = 139	*p* Value
Duration of disease, months, median (range)	10 (1 to 46)	4 (<1 to 15)	5 (1 to 14)	7 (<1 to 46)	<0.0001
Previous TKIs, *n* (%)					
0	41 (59)	23 (51)	8 (33)	72(52)	0.3
1	25 (36)	18 (40)	13 (54)	56 (40)	
>2	4 (5)	4 (9)	3 (12)	11 (8)	
Duration of treatment with current TKI, months, median (range)	8 (<1 to 18)	2 (<1 to 11)	4 (<1 to 11)	4.3 (<1 to 18)	<0.0001

TKIs, Tyrosine Kinase Inhibitors.

**Table 3 jcm-09-03417-t003:** Patients’ reported outcomes—additional questions.

Additional Questions	Imatinib *n* = 70	Dasatinib *n* = 45	Nilotinib *n* = 24	Total *n* = 139	*p* Value
Grade your health condition during the last 3 months *; mean (S.D.)	5 (1.2)	4.5 (1.3)	4.5 (1.5)	4.8 (1.3)	0.04
Was your QOL during the last 3 months affected by CML/treatment; *n* (%)					
Yes	26 (41)	26 (61)	17 (71)	69 (53)	0.02
No	38 (59)	17 (39)	7 (29)	62 (47)	
Symptoms since the current drug started					
Was your work function impaired?; *n* (%)					
Yes	22 (33)	24 (56)	12 (52)	58 (44)	0.04
No	45 (67)	19 (44)	11 (48)	75 (56)	
Was your home function impaired?; *n* (%)					
Yes	16 (24)	21 (47)	11 (46)	48 (35)	0.023
No	51 (76)	24 (53)	13 (54)	88 (65)	
Was your sexual function impaired?; *n* (%)					
Yes	25 (39)	17 (42)	9 (41)	51 (40)	0.94
No	39 (61)	23 (58)	13 (59)	75 (60)	
Were you followed at the cardiology/vascular/pulmonology/neurology clinic?; *n* (%)					
Yes	40 (61)	25 (57)	12 (52)	77 (58)	0.77
No	26 (39)	19 (43)	11 (48)	56 (42)	
Did you have difficulties in breathing due to fluids accumulation in your lungs?; *n* (%)					
Yes	8 (13)	9 (21)	4 (18)	21 (16)	0.5
No	36 (87)	34 (79)	18 (82)	108 (84)	
Were you admitted to the hospital in the last 6 months?; *n* (%)					
Yes	34 (49)	17 (38)	8 (23)	59 (42)	0.32
No	36 (51)	28 (62)	16 (67)	80 (58)	

* Health condition during the last month was graded between 0 and 6 (0—worst, 7—best). S.D., Standard Deviation; QOL, Quality of Life; CML, Chronic Myeloid Leukemia.

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
