# Peer review of "Health-Related Quality of Life in Patients with Chronic Myeloid Leukemia Treated with First- Versus Second-Generation Tyrosine Kinase Inhibitors"

_jcm, 2020, doi:10.3390/jcm9113417_

Round 1

Reviewer 1 Report

Shacham-Abulafia et al present an interesting “real-life” study on the quality of life (QOL) in Chronic Myeloid Leukemia (CML) patients under Tyrosine Kinase Inhibitors (TKIs) treatment. The authors report the results of three questionnaires completed from 193 patients receiving first and second generation TKIs. Among them, 139 cases receiving either imatinib (IM), dasatinib (DA) or nilotinib (NI) were included in the present study.

CML on IM were older and showed a longer duration of disease and treatment, compered to DA and NI treated that showed also a more severe symptomatic profile. Furthermore, patients on IM expressed a higher level of satisfaction from the treatment received.

The findings presented, showed as in the 15% of the Israelian CML patients, the subgroup treated with first generation TKIs reported a better QOL compared to second generation ones.

The issue is well introduced, the methodological approach is clearly described, the data are unambiguously presented and the conclusions are supported by the results observed. The manuscript is clear and results suitable for publication after minor revisions:

1) line 35: replace “EOTOS” with “EUTOS”.

2) line 71: “was” is reiterated.

3) line 99 (Table 1): specify the unit of measure of “Age”.

4) line 100 (Table 2): specify the unit of measure of “Duration of disease”.

5) line 100 (Table 2): specify the unit of measure of “Duration of treatment with current TKI”.

6) line 100 (Table 2): in the “Dasatinib duration of treatment” is “>” a typo?

Author Response

1) line 35: replace “EOTOS” with “EUTOS”.  

EOTOS was replaced to EUTOS

2) line 71: “was” is reiterated.

Corrected

3) line 99 (Table 1): specify the unit of measure of “Age”.

We added "years" to specify the measure of "Age".

4) line 100 (Table 2): specify the unit of measure of “Duration of disease”.

We added "months" to specify the measure of "Duration of disease".

5) line 100 (Table 2): specify the unit of measure of “Duration of treatment with current TKI”.

We added "months" to specify the measure of "Duration of treatment".

6) line 100 (Table 2): in the “Dasatinib duration of treatment” is “>” a typo?

We changed to <.

Reviewer 2 Report

In this manuscript, Shacham-Abulafia al. analyzed Quality of Life (QOL) perception in chronic myeloid leukemia (CML) patients treated with first and second generation Tyrosine Kinase Inhibitors. To assess QOL, they used a 30-item core questionnaire, a 24- item CML-specific questionnaire, both from the European Organization for Research and Treatment of Cancer (EORTC), and additional health-related items. In this way they interviewed about 15% of the Israeli CML patients, observing that mostly elderly patients were treated with imatinib and younger with dasatinib or nilotinib, and that not only age is associated with current TKI treatment, but it also predicts how patients perceive their QOL.

Overall, the manuscript is well-written and clear, and in my opinion it should be acceptable for publication in Journal of Clinical Medicine.

There are only minor comments:

  • L35: EUTOS for EOTOS
  • Along the text there are several double spaces.

Author Response

1) L35: replace “EOTOS” with “EUTOS”.  

EOTOS was replaced to EUTOS.

This manuscript is a resubmission of an earlier submission. The following is a list of the peer review reports and author responses from that submission.